# State of Charge Estimation Model Based on Genetic Algorithms and Multivariate Linear Regression with Applications in Electric Vehicles

**DOI:** 10.3390/s23062924

**Published:** 2023-03-08

**Authors:** Carlos Gustavo Manriquez-Padilla, Isaias Cueva-Perez, Aurelio Dominguez-Gonzalez, David Alejandro Elvira-Ortiz, Angel Perez-Cruz, Juan Jose Saucedo-Dorantes

**Affiliations:** Engineering Faculty, Campus San Juan del Río, Universidad Autónoma de Querétaro, Av. Río Moctezuma 249, San Juan del Río 76807, Querétaro, Mexico

**Keywords:** electric-vehicle, battery, state-of-charge, multivariate linear regression, genetic algorithms, estimation model

## Abstract

Nowadays, the use of renewable, green/eco-friendly technologies is attracting the attention of researchers, with a view to overcoming recent challenges that must be faced to guarantee the availability of Electric Vehicles (EVs). Therefore, this work proposes a methodology based on Genetic Algorithms (GA) and multivariate regression for estimating and modeling the State of Charge (SOC) in Electric Vehicles. Indeed, the proposal considers the continuous monitoring of six load-related variables that have an influence on the SOC (State of Charge), specifically, the vehicle acceleration, vehicle speed, battery bank temperature, motor RPM, motor current, and motor temperature. Thus, these measurements are evaluated in a structure comprised of a Genetic Algorithm and a multivariate regression model in order to find those relevant signals that better model the State of Charge, as well as the Root Mean Square Error (RMSE). The proposed approach is validated under a real set of data acquired from a self-assembly Electric Vehicle, and the obtained results show a maximum accuracy of approximately 95.5%; thus, this proposed method can be applied as a reliable diagnostic tool in the automotive industry.

## 1. Introduction

In recent decades, the demand for green/eco-friendly technologies has increased due to climate change concerns [1]. However, several issues must be addressed through the implementation of eco-friendly technology, and Electric Vehicles (EVs) have emerged and attracted the attention of the scientific community because they represent an alternative solution to those vehicles that work with an Internal Combustion Engine (ICE). In this regard, EVs are preferred to their counterparts because they offer a zero-emission alternative—such as in the case of Battery Electric Vehicles (BEVs)—due to the absence of an ICE [2,3]. Most EVs appear to be similar, but the main differences relate to the different battery technologies, where Lithium–Ion (Li–Ion) batteries stand out because of their high energy density, nominal voltage, cost, and long service life [4,5].

However, there are certain drawbacks and potential problems with EVs that automotive industry customers need to consider before considering buying one. For example, one of the main challenges for the massive adoption of EVs lies in their cost compared to an ICE vehicle, in particular, the high investment due to the cost of the battery pack, which represents one of the most expensive and critical components [6]. This is especially challenging in developing countries, where the commercialization of EVs has not been achieved on a relatively massive scale, and the electrical infrastructure is not prepared or has not been considered for modification for the adoption of EVs in comparison with developed countries such as China, the United States, and European countries [7]. Furthermore, in developing countries, factors such as income per capita or quality of life would not make it easy for a customer to consider acquiring an EV [8].

In the case of developing countries, the initial investment can be substantially reduced by adopting the vehicle conversion approach [9], which consists of taking a conventional ICE vehicle and turning it into an EV. This can be achieved with the acquisition, installation, and integration of components such as an electric motor, a motor controller, a battery pack, a battery management system (BMS), a charger unit, harnesses, digital foot throttles, vacuum pumps, and some other complementary modules that replace their conventional counterparts [10]. Even with the acknowledgment of this approach, other concerns arise regarding the efficiency and performance of the EV. To cite an example, an issue called “range anxiety” has appeared because of some problems related to the battery aspects of the vehicle, such as the distance range that can be covered with a single charge, charging time, battery replacement cost, and other limitations related to infrastructure [11] that do not appear in the case of conventional fuel vehicles.

Batteries with Li–Ion technology are widely used in EVs as their power source [12]; however, they are non-linear systems with high complexity. Their behavior has been reported to be mainly dependent on factors such as the State of Charge (SOC), temperature, and aging [13]. Therefore, it can be said that knowing or predicting the battery behavior is a key factor to achieving an efficient performance in electric vehicles, allowing the average driver’s trust in EVs to increase. However, SOC, being the most important parameter that describes the energy capacity of the batteries, cannot be measured directly with a sensor [14,15]. In this regard, Table 1 shows various methods that have been proposed by researchers for the estimation of the SOC in EVs, and they can be classified into different categories, such as conventional methods, model-based estimation methods, adaptive filtering methods, nonlinear observers, and learning algorithms [3].

Researchers have mainly used the pack voltage, current, and temperature in combination with different techniques that involve Artificial Intelligence (AI) and circuit modeling in order to estimate SOC. For instance, Liu et al. [1] developed a Thevenin model with variable parameters that are affected by the temperature. An Unscented Particle Filter (UPF) was used to estimate the SOC, where a Lithium battery was used to perform different charge–discharge tests at different temperatures between −10 °C and 45 °C under a controlled environment, obtaining good levels of accuracy when evaluating the fit of the battery model. In another example, He et al. [17] implemented an equivalent circuit model by means of using an Extended Kalman Filter and a parameter identification algorithm, which is based on adaptive Recursive Least Squares (RLS). The experiments were performed using the Dynamic Stress Test (DST) and the Urban Dynamometer Driving Schedule (UDDS) driving cycles at a fixed temperature of 25 °C within a controlled environment. Their proposed method worked well with both tests, indicating a high SOC estimation accuracy and robustness. In another work, Xiong et al. [18] used a Genetic Algorithm and Least Squares (GA-LS) method to perform the parameter identification in a two Lithium–Ion battery cell, in combination with an Unscented Kalman Filter method for the final SOC estimation. The experiments were performed under a controller environment using two temperatures: 10 °C and 25 °C. The authors performed UDDS and DST tests to simulate the current excitation to the cells. A maximum Root Mean Square Error (RMSE) of 0.85% was obtained when estimating the SOC.

In another example, Ali et al. [20] used the pack and voltage current in combination with the Lagrange multiplier method to identify the battery model parameters. An equivalent circuit model based on a first-order RC model was obtained in order to perform the SOC estimation. The voltage and current were monitored during charge and discharge tests performed under constant temperature and constant current conditions to estimate the circuit parameter values. The proposed technique had a maximum error below 1.5%. SOC estimation is also relevant in charging control, as miscalculations of this value could lead to potential failure of the battery. Therefore, Meng et al. implemented an equivalent circuit model in combination with an Extended Kalman Filter [21] that estimated the state of the battery and included the SOC value in a Model Predictive Charging Control. Simulation results showed good time performance in comparison with the traditional CC–CV (Constant Current–Constant Voltage) charging process. On the other hand, AI algorithms and machine learning techniques have also been considered to estimate the SOC in several applications; the most used and relevant techniques are stochastic Fuzzy Neural Networks, Adaptive Neuro-Fuzzy Inference System (ANFIS), Neural Networks such as Recurrent Neural Networks (RNN) and Fed-Forward Neural Networks (FNN), and Support Vector Machines (SVM), among others [22,23]. More recently, Deep Learning (DL) techniques have also been considered during the SOC estimation; in fact, Convolutional Neural Networks (CNN) [19] and autoencoders [24] are the most preferred. Nevertheless, although DL techniques can lead to the achievement of accurate results, their implementation can be associated with a high computational burden because a significant amount of data needs to be processed (i.e., CNN) and a priori knowledge is also mandatory to set specific values in the hyperparameters (i.e., autoencoder). For instance, Chitnis et al. [13] used an Artificial Neural Network (ANN) to obtain the SOC value. The voltage pack, voltage current, and integral of the current were used to train the ANN. A 144 V/216 Ah battery was used for the experiments, which consisted of a series of constant current charge and discharge tests. The neural network was used to estimate the SOC for Delhi Drive Cycles (DDC) as well as random drive cycles. Some authors have also implemented a combination of two or more methods to perform the SOC estimation. For instance, Liu et al. [4] used a Deep Belief Network (DBN) in combination with a Kalman filtering technique to estimate the SOC under dynamic conditions, where voltage, current, and temperature are used as the DBN inputs. The method was evaluated using DST and Randomized Battery Usage Data Set (RBUDS) tests to verify the accuracy of the proposed model. The experiments were performed on Lithium–Ion cells within a controlled environment chamber. The evaluated error values indicated that the accuracy of the method was good under dynamic conditions.

Although some of these works have presented improvements in the accuracy to model the battery or to estimate the SOC value, some limitations in these works have been observed: (i) They have not dealt with the fact that the battery discharge will not be the same with different kind of charges. For instance, the battery will not be discharged in the same way using an industrial motor or an EV motor. (ii) Some of these developments were also tested in laboratory environments where the battery banks were under controlled and fixed temperatures. That is, the batteries were tested by injecting voltage and current values using a test bench, but they were not validated in dynamic conditions within a field test. (iii) Little interest has been shown to consider the load-related variables as influential factors that affect the SOC, such as motor temperature, motor RPMs, vehicle speed, and vehicle acceleration.

Therefore, the contribution of this work lies in the proposal of a methodology to estimate and model the SOC based on Genetic Algorithms (GA) and multivariate linear regression with applications in electric vehicles. The proposed work initially considers the continuous monitoring of six load-related variables (the vehicle acceleration, vehicle speed, battery bank temperature, motor RPM, motor current, and motor temperature) to acquire the battery discharge behavior for posterior analysis. These variables have been shown to have a significant influence on the behavior of the battery bank discharge patterns when driving an Electric Vehicle following a defined trajectory. Subsequently, a GA-based structure is used in conjunction with Multivariate Linear Regression in order to identify and detect those representative load-related variables that minimize the Root Mean Squared Error (RMSE) during an optimization procedure where the SOC estimation model is performed. This proposal is evaluated under a real set of data acquired from a self-assembly EV, and the obtained results make this proposal feasible to be implemented as a reliable diagnostic tool in the automotive industry.

The proposed work is organized as follows. In Section 2, the genetic algorithm theory is explained in order to understand the treatment performed using the data obtained from the EV sensors. In Section 3, the methodology that is used to obtain the final SOC model estimation is included, and the experimental data are explained. Section 4 presents the experimental setup and the description of the material resources used in this present work. In Section 5, the results are presented and discussed to explain the key parameters selected to perform the final SOC model estimation, which is followed by the concluding remarks in Section 6.

## 2. Genetic Algorithms

This section presents the general aspects to be considered for implementing Genetic Algorithms (GA); for example, they are well known as stochastic algorithms used to solve complex, non-linear problems and/or those problems whose solution varies over time [25]. Figure 1 shows the flow chart of the operation of a simple GA structure.

In general terms, the algorithm imitates the mechanism of natural evolution and uses a set of possible solutions (population); to be concise, the implementation of GA starts with a randomly generated population, where a Gaussian random distribution allows one to improve the variability. Then, each solution corresponds to a chromosome representing a gene; the selection of the best chromosomes is performed by using selection operators such as Roulette Wheel, Boltzmann, and Tournament, among others [26].

Subsequently, following a fitness function, the GA rates the chromosomes, aiming to determine the best solution; once the best solution is chosen, crossover and mutation are performed to generate a new population composed of chromosomes that better fit the final solution, which solves the problem. The crossover operator imitates biology and leads the GA to two possible solutions, that are combined (parent solution) and thus there are two new solutions (children solutions) [27]. Common crossover techniques are single-point crossover, uniform crossover, cycle crossover, and multi-point crossover, among others [28]. Similarly, the mutation maintains diversity in the solutions by randomly modifying a gene from a chromosome generated by the crossover operator; thus, it leads the GA to find a local solution instead of a global solution. In this sense, common mutation techniques are power mutation, uniform, non-uniform, and Gaussian, among others [29]. Finally, the procedure is repeated until the stop criterion is reached; that is, the number of generations and/or minimization/maximization of the fitness function [30].

Common applications of GA, as reported in the literature, are those related to the solution of problems where parameter optimization is carried out, particularly in applications associated with multiparameter function optimization. This is because the strength of GA, compared to other optimization strategies, lies in the fact that genetic algorithms can optimize many parameters simultaneously [31].

## 3. Methodology

The proposed methodology for estimating the State of Charge (SOC) in Electric Vehicles (EVs) is composed of five stages, as depicted in Figure 2. As observed, the method includes the following: (i) the Electric Vehicle, (ii) data monitoring, (iii) data modeling, (iv) optimal predictors selection, and (v) SOC estimation. The overall process is developed with the aim of obtaining a mathematical model that allows one to predict the SOC of a battery bank in an electric vehicle in order to provide information regarding the autonomy of the vehicle. Every step is described in detail next.

Step (i) Electric vehicle: First, it should be highlighted that several tests were performed on a laboratory self-assembly Electric Vehicle, where two different conditions were studied. The experimental tests considered the assessment of the battery bank discharge behavior under two different transmission shifts (second and third). 

Step (ii) Data monitoring: The acquisition of six significant physical magnitudes was carried out to monitor the battery bank behavior during its discharge in driving tests when the transmission is fixed in two different shifts, second and third. The acquired signals were vehicle acceleration, vehicle speed, battery bank temperature, motor RPM, motor current, and motor temperature; these magnitudes are recommended to be used in the estimation of SOC because they provide significant information. Thus, during the driving test, the signals were continuously measured, and the data were stored on a removable SD card. In this way, it was possible to extracting live information from the BMS. The values for the six physical variables acquired were used without any additional data pre-processing, as the measurement equipment already delivers the real values for these physical magnitudes.

Step (iii) Data modeling: Once the data monitoring was performed, it was necessary to propose a model that describes the discharge behavior of the battery pack. In this work, it is considered that the battery pack SOC can be described as a linear combination of all or some of the physical variables acquired in step (ii). Thus, the result is the model presented in Equation (1):(1)$$\mathrm{SOC}=\beta_{0}+\beta_{1}x_{1}+\beta_{2}x_{2}+\beta_{3}x_{3}+\beta_{4}x_{4}+\beta_{5}x_{5}+\beta_{6}x_{6}$$
where $x_{1},\;x_{2},\ldots ,\;x_{n}$ are the physical variables measured at step (ii) and $\beta_{0},\;\beta_{1},\ldots ,\;\beta_{n}$ are the coefficients estimated using multivariate linear regression. In Table 2 are summarized the collected data and the assigned variable.

Step (iv) Optimal predictors selection: Although six different measurements were acquired, it is possible that some of the physical variables measured provide more significant information regarding the SOC of the battery pack, otherwise it is also possible that non-useful information and/or correlated information is depicted by these considered variables. In this regard, the accuracy during the SOC estimation may be compromised depending on whether or not the SOC model is based on those non-relevant variables. Accordingly, the implementation of the GA leads to the optimized selection of the variables, which can improve the SOC modeling in terms of multivariate regression; thus, the optimal predictor selection was carried out as follows:(1)The first and second steps define whether the original six variables are considered or not; for example, in the first-round the vehicle speed, vehicle acceleration, and motor RPMs are selected, which are then evaluated according to a specific model; for this proposal, the evaluation is achieved through Equation (1). It is important to mention that this first selection is randomly carried out.(2)The selection of the best chromosomes, which is accomplished by Roulette Wheel, leads to the problem solution; in fact, the solution is performed by selecting the predictors/variables that produce a minimization in the fitness function, which is in terms of the Root Mean Square Error (RMSE), as shown in Equation (2):
(2)$$f_{c}=\min\left(\mathrm{RMSE}\right)$$
where the RMSE value is a standard way to measure the error of a model in predicting quantitative data. The formula for the RMSE calculation is presented in Equation (3):(3)$$\mathrm{RMSE}=\sqrt{\overset{n}{\underset{i=1}{\sum }}\frac{{\left({\hat{y}}_{i}-y_{i}\right)}^{2}}{n}}$$
where n is the number of observations, ${\hat{y}}_{i}$ represents the predicted values, and $y_{i}$ represents the observed values.

(3)To prevent stagnation, a mutation probability of 25 is considered; this step is implemented by generating a random number between 0 and 1, and the mutation is only applied if the result is lower than 0.25. In consequence, the value of one of the chromosomes is arbitrarily changed.(4)Finally, the process is repeated until one of the stop criteria is reached, which include reaching the maximum number of generations or by minimizing the fitness function.

Step (v) SOC estimation: Because the optimal predictors have been previously determined, they are selected to perform a final multivariate linear regression. With this regression, a linear model is approximated that describes the SOC of the battery pack.

Step (vi) Validation: Finally, another test is performed in order to validate the model generated in the previous steps. This test is conducted in the second transmission shift.

## 4. Experimental Setup

This section presents the materials and equipment used in the experimental tests. The test conditions are also explained.

The self-assembly EV used for the experiments is shown in Figure 3; this vehicle is a conventional Hyundai Atos that was converted into an Electric Vehicle in the Faculty of Engineering of Universidad Autonoma de Queretaro (UAQ). The EV is powered by a battery bank that consists of 32 Lithium cells (LFMP100AHX) connected in a series configuration with a nominal capacity of 100 Ah, a LiFeMnPO4 chemistry, and a nominal voltage of 3.3 V. Therefore, the approximate nominal pack voltage is 105.6 V when completely charged.

The EV conversion was performed by replacing the conventional ICE motor with an AC-35 26.25 Induction Motor (B-Face. 1 1/8″ keyed shaft), which produces 63 hp (46.97 kW) at 2900 RPM while delivering 129 ft-lbs. of torque at 96 V and 650 A. This motor works in combination with a Curtis 1238E-7621 induction motor controller. The transmission, brake, and direction systems were not altered during the vehicle conversion. An Orion Battery Management System version 2 was also installed to monitor the battery cells and to coordinate and manage diverse tasks in the EV, such as charging, discharging, and cell-balancing. This BMS is CAN enabled, and therefore it has the capability to send and receive information to other modules of the system, such as controllers, loggers, and chargers via the CAN protocol.

All the experiments were performed in San Juan del Rio Campus of the Faculty of Engineering of UAQ on a fixed track within the campus. Five tests were performed for each transmission shift. Each test consisted of traveling within the track for a total distance of 15 km. After each test, the batteries were charged up to 95% of SOC to provide a suitable SOC range to perform the model estimation. During each test, the EV was driven, maintaining an approximately constant speed. For the second shift tests, a 15 km/h speed was used, and for the third shift tests, a speed of 25 km/h was used. 

The test steps are presented as follows:

Step (i) EV charge: The battery pack was charged using an ELCON HK-J-H132-32 charger controlled via a CAN bus by the Orion BMS to ensure that each test started with a 95% SOC. 

Step (ii) Measurement configuration: A 1313 Handheld Programmer for Curtis was used to acquire five of the six variables: the vehicle acceleration, vehicle speed, motor RPM, motor current, and motor temperature. The remaining variable (battery bank temperature) was acquired with an Orion BMS v2. The handheld was connected to the motor controller via USB before starting each driving cycle and was then configured to acquire those signals during the complete trip. A similar procedure was carried out for the Orion BMS to acquire the bank temperature. The BMS was configured to acquire that variable before each trip. All six variables were acquired during the complete trip. The sample rate for these measurements was two samples per second for all the variables, except for the bank temperature, whose sample time was 100 milliseconds.

Step (iii) Trip and data acquisition: The data acquisition was started, and the EV travelled for 15 km, trying to keep a constant speed, which depended on the shift used for that specific test.

Step (iv) End of trip: When the EV had traveled for 15 km, the test ended and the data acquisition was stopped. After this, the data in the handheld programmer was imported to a PC to perform data curation and processing.

## 5. Results and Discussion

During the optimization procedure, a Genetic Algorithm (GA) structure is used to find those variables (inputs) that better fit with a given model response (output); thus, in the optimization process the GA performs a linear regression to find a model between inputs and outputs, and the GA fitness function aims to minimize the Root Mean Squared Error (RMSE). Furthermore, with the predictions obtained from the genetic algorithm, a final multivariate linear regression model is fed, which allows one to obtain a numerical model that estimates the SOC value of the battery bank as a function of the selected input variables.

To apply the previously described methodology, the six input variables listed below were selected: (i) motor current, (ii) motor RPMs, (iii) vehicle acceleration, (iv) vehicle speed, (v) motor temperature, and (vi) average battery temperature. The variables mentioned above were acquired during the 15 km run. As an example of the behavior of the six variables monitored during the experimental tests, Figure 4 shows the signals acquired during the first 600 s of one of the tests carried out selecting the second transmission shift.

The proposed method was evaluated under a series of experimental tests considering two transmission shifts, where five repetitions were performed for the second and third shifts, iteratively. Thus, a total of ten experiments were carried out, and nine of these tests were considered for training purposes (four from the second shift and five from the third shift), and the remaining were used for testing purposes (the fifth test of the second shift). Subsequently, the data acquired in the experimental tests were fed into the GA, in conjunction with a multivariate linear regression in order to obtain an SOC model that best minimized the Root Mean Square Error (RMSE). The lower the RMSE value, the greater the influence of the selected input variables on the output variable.

For the experimental test carried out in the second transmission shift, the results matrix shown in Table 3 and Table 4 were obtained; to demonstrate the accuracy and reliability of the proposed approach, an ANOVA test was performed, and the most significant results are also shown in Table 3 and Table 4. According to the *p*-value, the selected variables are those relevant during the SOC modeling estimation, and the *R*-squared proves that the accuracy can reach maximum values of around 95.5%.

As can be seen in Table 3, the lowest RMSE value was obtained in experimental test 2, and the highest value was obtained in experimental test 1. Figure 5 shows the behavior of the RMSE value through the 50 generations of the GA for the two cases. In a similar manner, for the tests carried out in the third transmission shift, the results matrix shown in Table 4 was obtained.

For this case, the lowest RMSE value was obtained in test 5 and the highest value was obtained in test 1. The evolution of these values through the GA generations is shown in Figure 6.

Once the data from the genetic algorithm had been obtained, the mathematical model that governs the behavior of the output variable was estimated through multivariate linear regression. For this case, the output variable was the SOC of our vehicle’s battery pack. For illustrative purposes, Figure 7 shows the estimation of the SOC for the tests described in Figure 5 corresponding to the tests carried out at the second transmission shift.

On the other hand, the SOC obtained for the third transmission shift tests described in Figure 6 are shown in Figure 8.

As can be seen in the results obtained, the variables that have the greatest influence in general on the SOC of a battery pack are motor temperature and motor RPM. Additionally, it was found that, at low speeds, the monitoring of average battery temperature contributes to a great extent to determining the SOC; similarly, for high speeds, the variable that contributes to this is the speed of the vehicle. This is because, at low speeds, due to the design of the vehicle, the battery bank experiences an increase in temperature because the air current from the outside of the vehicle is not large enough. In the case of high speeds, the airflow from outside has an effect on the battery bank and helps to control their temperature; however, the torque required to move the vehicle at said speeds is greater, which has an impact on the amount of energy that the electric motor demands from the battery bank.

Figure 9 shows the comparison between the SOC estimated using the previously described methodology and the SOC monitored in real-time during the experimental tests. As observed in Figure 9, both values of SOC, real and estimated, have a similar behavior trend over time; therefore, it can be said that the methodology presented in this paper is useful to model the real behavior of the SOC of a battery bank corresponding to an electric vehicle. Although the trend of the SOC is similar to that measured, it can be seen that there is still an error between them. This error may be reduced by filtering the signals; however, in the case of online applications, it may be improved by resampling the estimated SOC. Unlike the methodologies to estimate the SOC reported in the literature, the methodology presented in this paper considers various non-linear parameters found while driving an electric vehicle following a path implemented over real traffic patterns and road characteristics.

## 6. Conclusions

Electric mobility seems to be an important factor towards reducing the amount of greenhouse gas emissions. However, there are some challenges that must be faced to ensure the robustness and reliability of electric vehicles. One of the most critical factors is the energy stored in the battery pack, because the autonomy of the EV depends on it. In this sense, the development of strategies for the accurate estimation of the SOC of the battery pack is essential in order to achieve these goals of robustness and reliability. Most of the reported works so far perform SOC estimation using the current and voltage from the batteries; notwithstanding, there are some other factors, such as temperature, that may affect the rate of discharge of the batteries. Additionally, the driving conditions may vary from one use to another; therefore, it is important to consider parameters such as the motor and vehicle speed, and acceleration. Every EV is always integrated with a BMS that provides a significant amount of information regarding different physical variables related to the operating conditions of the vehicle, and specifically of the battery pack. However, not all the information results are relevant for the estimation of the SOC of the batteries. In this sense, the proposed methodology proved that a GA is an effective tool to determine whether or not a parameter is relevant for the SOC estimation. Therefore, the obtained results prove that motor RPM, motor temperature, battery temperature, and vehicle speed are parameters that must be included in the models to determine the SOC of the battery pack in order to obtain a more adequate estimation. With the use of these variables, it is possible to predict the discharge pattern that will be experienced by the batteries using a simple approach with a multi-variate linear regression model. This way it is possible to provide the users with information regarding when it is necessary to recharge the batteries so it can be scheduled. This method aims to be a tool to cope with the actual BMS to improve the information that is provided to the final users.

## Figures and Tables

**Figure 1 sensors-23-02924-f001:**
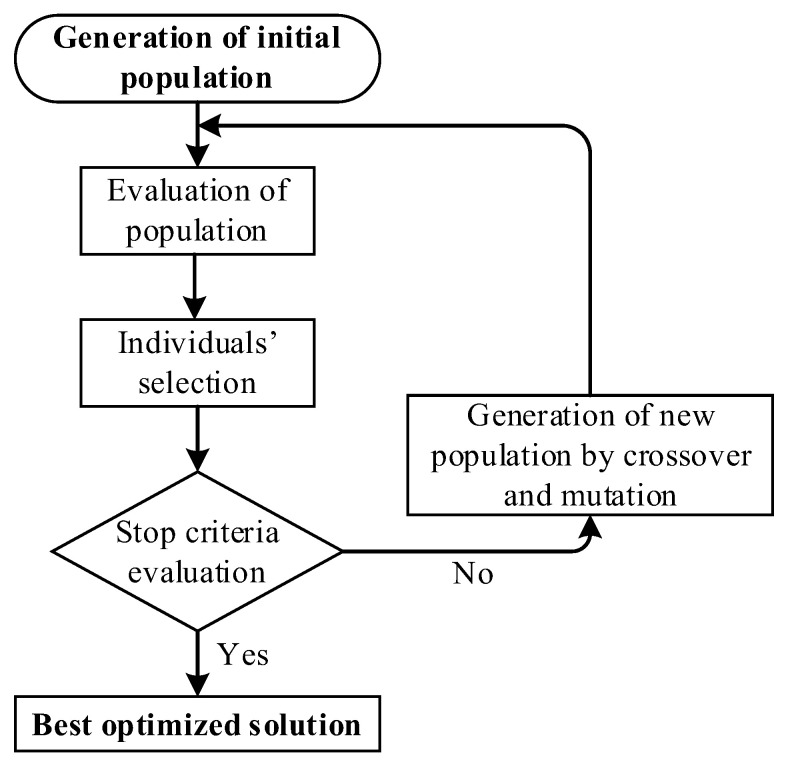
General flow chart of a simple GA structure that is commonly used in engineering applications.

**Figure 2 sensors-23-02924-f002:**
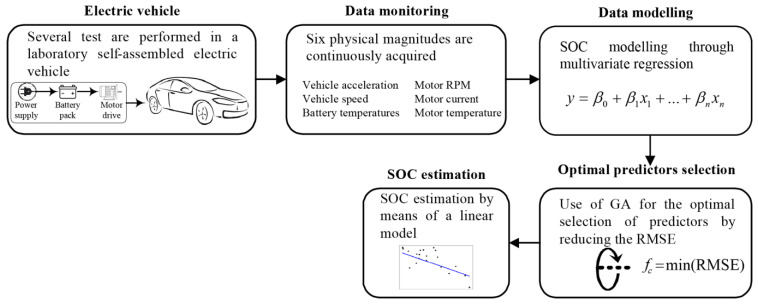
Flow chart of the proposed methodology based on a multivariate regression approach and GA for estimating the SOC of an electric vehicle.

**Figure 3 sensors-23-02924-f003:**
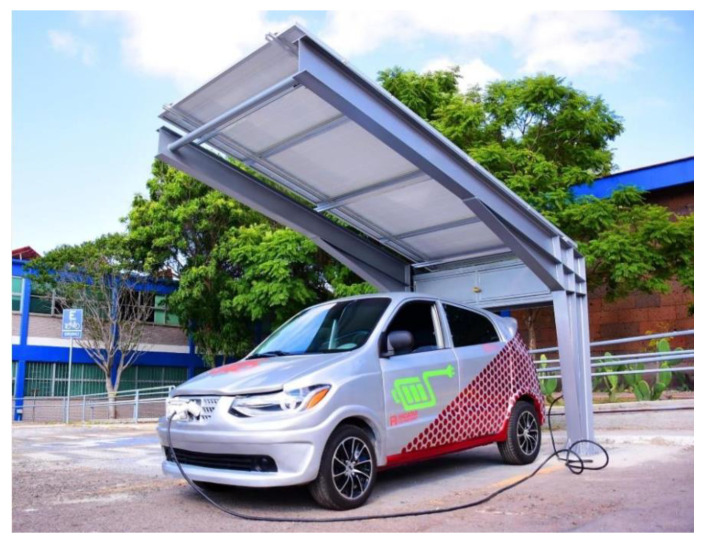
Self-assembly EV used for experimental tests for the estimation of SOC by means of GA and multivariate regression.

**Figure 4 sensors-23-02924-f004:**
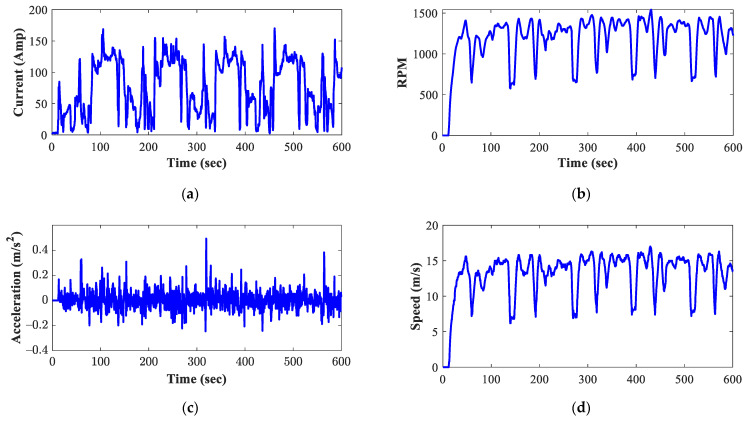
Set of signals acquired during the experimental tests: (**a**) motor current, (**b**) motor RPMs, (**c**) vehicle acceleration, (**d**) vehicle speed, (**e**) motor temperature, and (**f**) average battery temperature.

**Figure 5 sensors-23-02924-f005:**
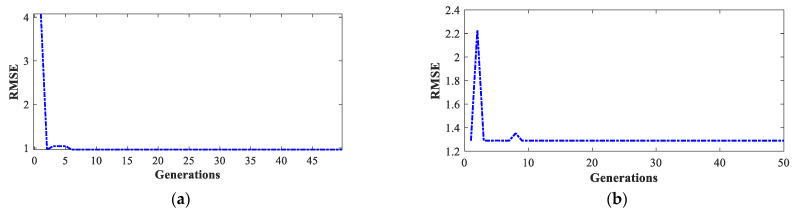
Evolution of the RMSE value through the generations for the second transmission shift tests: (**a**) test 2 (test with the lowest RMSE value) and (**b**) test 1 (test with the highest RMSE value).

**Figure 6 sensors-23-02924-f006:**
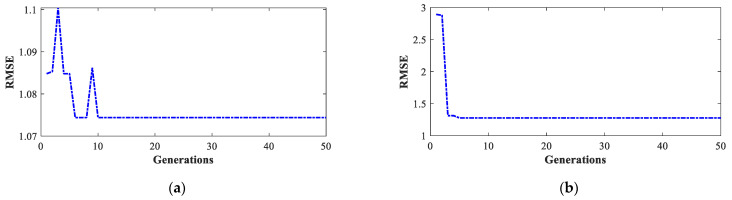
Evolution of the RMSE value through the generations for the third transmission shift tests: (**a**) test 5 (test with the lowest RMSE value) and (**b**) test 1 (test with the highest RMSE value).

**Figure 7 sensors-23-02924-f007:**
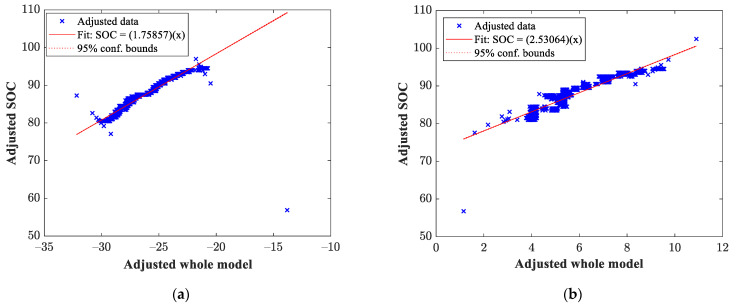
Estimated SOC obtained in the test using the second transmission shift: (**a**) test 2 (test with the lowest RMSE value) and (**b**) test 1 (test with the highest RMSE value).

**Figure 8 sensors-23-02924-f008:**
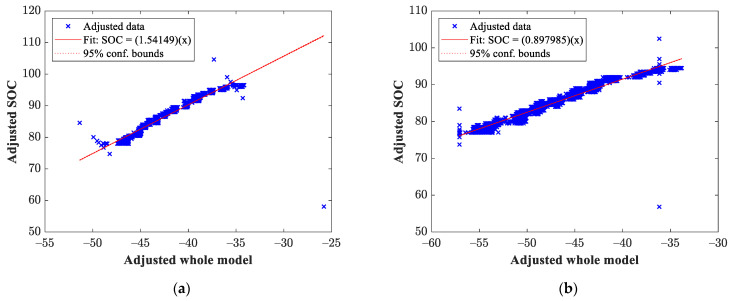
Estimated SOC obtained in the test using the third transmission shift results: (**a**) test 5 (test with the lowest RMSE value) and (**b**) test 1 (test with the highest RMSE value).

**Figure 9 sensors-23-02924-f009:**
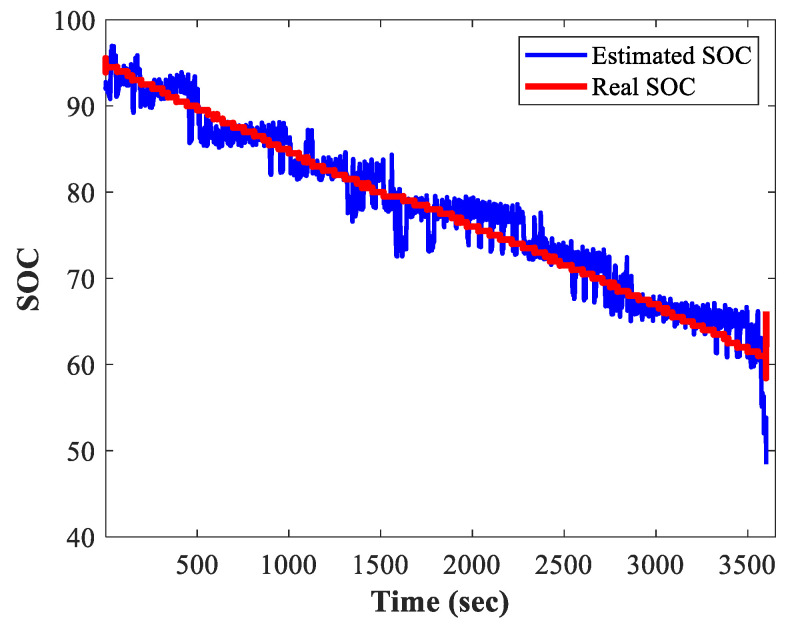
Visual representation of the SOC estimated by the proposed methodology (blue) and the real-time SOC measured during an experimental test (red).

**Table 1 sensors-23-02924-t001:** Classification of SOC estimation methods.

SOC Estimation Method	Examples	References
Conventional methods	Coulomb counting Open Circuit Voltage (OCV) Electromotive Force (EMF) Internal resistance	[16]
Model-based methods	Equivalent Circuit Models (ECM) Electrochemical Models (EM) Electrochemical Impedance Spectroscopy Models (EISM) Reduced-Order Models (ROM)	[17]
Adaptive filter algorithms	Kalman Filters (KF) Extended Kalman Filters (EKF) Unscented Kalman Filters (UKF) Adaptive Unscented Kalman Filters (AUKF)	[18]
Nonlinear observers	Sliding Mode Observers (SMO) Adaptive Switching Gain Sliding Mode Observer (ASGSMO) Nonlinear Observer (NLO) Proportional Integral Observer (PIO)	[1]
Learning algorithms	Neural Networks (NN) Artificial Neural Networks (ANN) Feedforward Neural Networks (FNN) Support Vector Machines (SVM) Fuzzy Logic (FL) Genetic Algorithms (GA)	[19]

**Table 2 sensors-23-02924-t002:** Collected data for estimating the SOC model.

Physical Magnitude	Assigned Variable
Vehicle acceleration	$x_{1}$
Vehicle speed	$x_{2}$
Battery bank temperature	$x_{3}$
Motor RPM	$x_{4}$
Motor current	$x_{5}$
Motor temperature	$x_{6}$

**Table 3 sensors-23-02924-t003:** Variables that best minimize the RMSE for the second transmission shift.

Experimental Test	Variables That Better Minimize the RMSE	Achieved RMSE Value	*R*-Squared	*p*-Value
1	Vehicle acceleration, Vehicle speed, Motor temperature, Average battery temperature.	1.2901	0.896	9.52 × 10^−8^
2	Motor current, Motor RPM, Motor temperature, Average battery temperature.	0.9609	0.944	0
3	Motor RPM, Motor temperature, Average battery temperature.	1.1781	0.922	0
4	Motor current, Motor RPM, Motor temperature, Average battery temperature	1.1009	0.925	0

**Table 4 sensors-23-02924-t004:** Variables that best minimize the RMSE for the third transmission shift.

Experimental Test	Variables That Better Minimize the RMSE	Achieved RMSE Value	*R*-Squared	*p*-Value
1	Motor RPM Vehicle speed Motor temperature Average battery temperature	1.2770	0.949	0
2	Motor current Motor RPM Vehicle speed Motor temperature	1.2025	0.948	0
3	Motor current Vehicle speed Motor temperature Average battery temperature	1.1355	0.955	0
4	Motor current Motor RPM Vehicle speed Motor temperature	1.1756	0.946	0
5	Motor current Motor RPM Vehicle speed Motor temperature	1.0744	0.955	0

## Data Availability

The data presented in this study are available on request from the corresponding author. The data are not publicly available due to the nature of the project.

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
