# Peer review of "State of Charge Estimation Model Based on Genetic Algorithms and Multivariate Linear Regression with Applications in Electric Vehicles"

_sensors, 2023, doi:10.3390/s23062924_

Round 1

Reviewer 1 Report

Paper has following issue. Addressing them is essential

What does the author mean by  six meaningful load-related variables which have an influence that affects the SOC must be given in the abstract.

 Very poor grammatical orientation   The specifications regarding self assembled EV in lab need to be explained for generalized usage   in abstract line no.14 unnecessary use of 'In' and 'Is' for sentence formation.   The abbreviation of SOC isn't mentioned at the start of the paper.   line no. 233 are RMSE function to be describe in detail   Authors can compare the database,methodology and results with laboratory tests as well.(SOC comparison) Authors may review artificial intelligence (other than GA) in further attempts. Authors can elaborate about the optimization procedure for Genetic algorithm (GA) structure(Simulation model can be included).   line no339,340and figure no 6 '#'  is unnecessarily used to mentioned result

Author Response

We, the authors, want to thank the reviewers and Editors for their useful comments to improve the manuscript. We have tried our best to fulfill such valuable comments. All changes made to the manuscript are highlighted in yellow.

Reviewer 2 Report

This paper presents an interesting research work about battery SOC estimation. 

1) battery SOC estimation is a widely studied research topic, however, it is not an independent research topic in reality. For example, the estimated SOC should be used in energy management strategy, energy storage control, diagnosis etc. Therefore, the introduction should be improved with more recent research work such as Meng, Jianwen, Meiling Yue, and Demba Diallo. "Nonlinear extension of battery constrained predictive charging control with transmission of Jacobian matrix." International Journal of Electrical Power & Energy Systems 146 (2023): 108762.

2) the discussion about the precision of the SOC estimation results should be improved. The figure 9 gives the soc estimation results, however, more analysis about the estimation precision should be added.

Author Response

(The authors gave the same response as above.)

Reviewer 3 Report

1. Please report quantitative results in the Abstract section. 

2. More critical literature analysis is needed. For instance, a table can be used to summarize the existing methods of SOC estimation mentioned in Lines 72 to 111. 

3. There are many sophisticated methods such as deep learning have been used for SOC estimation. Why the authors choose two relatively classical methods of genetic algorithm and multivariate linear regression to solve this problem? What are the strengths of proposed method as compared to deep learning based method? Please justify.

4. Section 2 Genetic algorithm is quite lengthy. Please reduce the lengthy and make it more concise 

5. Figure 1 is misleading. Crossover and mutation should be presented in separate blocks. Furthermore, selection process should appear after mutation  and before generating new population.

6. Lines 189 and 190 - The sequences of (iii) and (iv) are not consistent that those in Figure 2.

7. Data pre-processing are missing in Step III. Please elaborate the efforts made in data pre-processing.

8. Authors also should be more specific on the input data used to predict the SOC and this needs to be reflected in Eq. (1).

9. More details are needed in Step (iv). For instance, what are the decision variables to be optimized by GA in this case? How do these decision variables to be encoded into each chromosome? All these information should be presented in more details manner.

10. What are the data have been collected to estimate SOC? Please present these data. Authors also need to mention how the data split for training, validation and testing purposes.

11. There are no comparison between the proposed method and existing methods for estimating SOC. The strengths of proposed method cannot be validated.

Author Response

(The authors gave the same response as above.)

Round 2

Reviewer 2 Report

Thé questions have been answered, the paper can be accepted.

Author Response

We, the authors, want to thank the reviewers and Editors for their useful comments to improve the manuscript. We have tried our best to fulfill such valuable comments.

The manuscript has been improved after the review process.

Reviewer 3 Report

Thank you for taking proper actions to address each comment provided. I am quite satisfied with the respond provided. The only issue is with the flowchart of GA in Figure 1. After generating new population, fitness evaluation needs to be performed before selection operator is used to select next generation. Furthermore, the "Yes" and "No" labels are missing when checking the stopping criteria. Please revise the flowchart again.

Author Response

(The authors gave the same response as above.)
